# CAncer bioMarker Prediction Pipeline (CAMPP)—A standardized framework for the analysis of quantitative biological data

**Thilde Terkelsen[1], Anders Krogh[2], Elena Papaleo[1,3]***

**1** Computational Biology Laboratory, Danish Cancer Society Research Center and Center for Autophagy, Recycling and Disease, Copenhagen, Denmark, **2** Unit of Computational and RNA biology, Department of Biology, University of Copenhagen, Copenhagen Denmark, **3** Translational Disease System Biology, Faculty of Health and Medical Science, Novo Nordisk Foundation Center for Protein Research, University of Copenhagen, Copenhagen, Denmark

* elenap@cancer.dk

**Data Availability Statement:** All data are available without restriction in GitHub: https://github.com/ELELAB/Cancer-bioMarker-Prediction-Pipeline-CAMPP.

## Abstract

With the improvement of -omics and next-generation sequencing (NGS) methodologies, along with the lowered cost of generating these types of data, the analysis of high-through-put biological data has become standard both for forming and testing biomedical hypotheses. Our knowledge of how to normalize datasets to remove latent undesirable variances has grown extensively, making for standardized data that are easily compared between studies. Here we present the CAncer bioMarker Prediction Pipeline (CAMPP), an open-source R-based wrapper (https://github.com/ELELAB/CAncer-bioMarker-Prediction-Pipeline -CAMPP) intended to aid bioinformatic software-users with data analyses. CAMPP is called from a terminal command line and is supported by a user-friendly manual. The pipeline may be run on a local computer and requires little or no knowledge of programming. To avoid issues relating to R-package updates, a renv .lock file is provided to ensure R-package stability. Data-management includes missing value imputation, data normalization, and distributional checks. CAMPP performs (I) k-means clustering, (II) differential expression/abundance analysis, (III) elastic-net regression, (IV) correlation and co-expression network analyses, (V) survival analysis, and (VI) protein-protein/miRNA-gene interaction networks. The pipeline returns tabular files and graphical representations of the results. We hope that CAMPP will assist in streamlining bioinformatic analysis of quantitative biological data, whilst ensuring an appropriate bio-statistical framework.

## Introduction

The availability of sensitive and specific biomarkers for disease diagnosis, prognosis, and monitoring, is an attractive alternative to many of the current methods in use. The presence and levels of certain tissue-derived molecular markers can help distinguish subtypes in heterogeneous diseases such as cancer [1,2]. Biomarkers may also be predictive of patient outcome and responsiveness to treatment [3,4]. Alas, pinpointing robust cancer biomarkers may be a

**Funding:** EP received funds from Innovationsfonden (grant number 5189-00052B), Danmarks Grundforskningsfond (grant number DNRF125), and LEO foundation exploratory grant (grant number LF17006). The funders had no role in study design, data collection and analysis, decision to publish, or preparation of the manuscript.

**Competing interests:** The authors have declared that no competing interests exist.

challenging endeavor. In a review from 2014, Yotsukura and Mamitsuka [5] showed that out of 7720 publications on biomarkers usage, only 407 of these were patented, and none had obtained FDA approval [5]. One of the main limitations of biomarker research often relates to small sample size, yielding over-fitted and unreproducible results [6,7]. Other pitfalls include a lack of standardized data curation [8], inappropriate statistical analysis, and lack of validation [6,9,10]. Evaluation of marker specificity and sensitivity is pivotal as most cancer biomarkers have high false-positive rates since a range of non-cancerous events may cause changes in levels of specific biomolecules. Despite these limitations, macromolecular markers remain a promising strategy for diagnosis and subtyping of cancer, as well as other diseases. With the advancements in the field of high throughput data and the increased attention to data normalization and statistical modeling, some drawbacks of biomarker mining may be overcome in the foreseeable future [11].

Central to the identification of novel disease markers is the bioinformatic analysis of high throughput biological data [12–16]. By applying different statistical tests and machine learning approaches, researchers can go from large datasets with quantitative measurements to a few biomolecules of interest [17–21]. Ideally, biomarker studies of the same disease should be reasonably comparable; however, discrepancies of results are not at all uncommon. While some of these differences arise from variances in study design and experimental procedures, a significant proportion is due to alternating and sometimes inappropriate data normalization and bioinformatic analysis pipeline [22–24]. Standardizing the framework for the detection of biomarker candidates both in the wet lab [25], as well as the dry lab, should enable researchers to more directly compare results across different studies and starting materials [24,26].

We here illustrate The CAncer bioMarker Prediction Pipeline (CAMPP), which is an R-based command-line wrapper for the analysis of high throughput data. The intention behind CAMPP is to provide bioinformatic software-users with a standardized way of screening for potential disease markers, and other biomolecules of interest, prior to potential experimental validation. We have aimed for a pipeline that is intuitive and well-documented, as is good practice [27].

## Design and implementation

### Requirements

The main prerequisite for CAMPP is R and Rstudio. Furthermore, Macbook users must have Xcode installed [https://developer.apple.com/xcode/], while windows users must ensure they have some equivalent of command-line tools.

Currently, CAMPP is a command-line tool run using Rscript, which is automatically acquired when installing R and Rstudio (most likely path; */usr/local/bin/Rscript)*. Minimum requirements; *R version 3.5.1* and *Rstudio version 1.1.463*.

### Set-up and package stability

The first time the pipeline is run, all R-packages and dependencies are automatically checked and installed. For installation, a default CRAN mirror is employed; however, this may easily be changed to better fit user location (https://cran.r-project.org/mirrors.html).

*Renv*: As updates of R-packages are regular, there is a chance that the code will break, and we therefore also provide the user with the option to use an R-package "freeze", in the form of a *renv* [28]. To use the *renv* packages, the user must run CAMPP with the argument flag, *-e*, set to "stable".

The CAMPPFunctions.R contains custom functions sourced by the CAMPP.R script (main script), and, as such, these two scripts must be located in the same working directory.

CAMPP may be used to perform a variety of analyses, including preliminary data management, summarized in Table 1 and Fig 1A below. As the analyses with CAMPP are standardized, the pipeline accepts a range of biological datasets from different high throughput sequencing platforms.

## User input

The user must provide; (I) a matrix with expression/abundance values, (II) a metadata file which should contain at least two columns, a column with sample ids, matching the column names in the expression/abundance file and a column specifying which group (disease state, treatment etc.) a given sample belongs to and (III) the user has to specify which type (variant) of data is provided for analysis, current options are; a*rray*, *seq*, *ms or other*.

## Value imputation and pseudocounts

If the input data contains missing expression values, these will automatically be imputed using Local Least Squares Imputation, LLSI [29], or K-Nearest Neighbor Imputation, KNNI [30]. The default method is LLSI, as this type of imputation has been shown to perform better on expression data [31].

## Data normalization, filtering, and transformation

For RNA sequencing data (*seq*) variables with low counts over all groups (tissue, treatment) are filtered out, library sizes are scaled (weighted trimmed mean of M-values, TMM) [32], and data are voom-transformed [17]. For microarray data, (*array*) data are log2-transformed and either quantile normalized [32,33] or standardized using mean or median [34]. Mass spectrometry data (*ms*), are log-transformed (log2, log10, or logit) and standardized using mean or median [34].

**Table 1. Table summarizing preliminary data management and analyses implemented in CAMPP, along with specific methods and underlying R-packages.**

|  | Type | R-package |
|---|---|---|
| **Preliminary Data Management** |  |  |
| Missing Value Imputation | K-nearest neighbor | impute [30] |
| Data Normalization | Normalize between arrays and mean/median centering | R Base |
| Data Transformation | log2, log10, logit, voom | R Base |
| Data Distribution Fitting | Normal, Weibull, Lognormal, Gamma, Poisson, Binomial | fitdistrplus [35] |
| **Analysis** |  |  |
| K-means Clustering | Hartigan and Wong algorithm, BIC | R Base, mclust [36] |
| Differential Expression/Abundance Analysis | Empirical Bayes Framework for Linear Models | edgeR [32], limma [37], sva [41] |
| LASSO / Elastic-Net Regression | Group LASSO/EN—multinomial | glmnet [42], pROC [43] |
| Co-Expression/Abundance Correlations | WGCNA, Pearson, Spearman | WGCNA [44], R Base |
| Survival Analysis | Cox Proportional Hazard Regression | survcomp [45] |
| Interaction Networks | STRING [47], miRTarBase [48] and TargetScan [49] | multiMiR [50] |

A

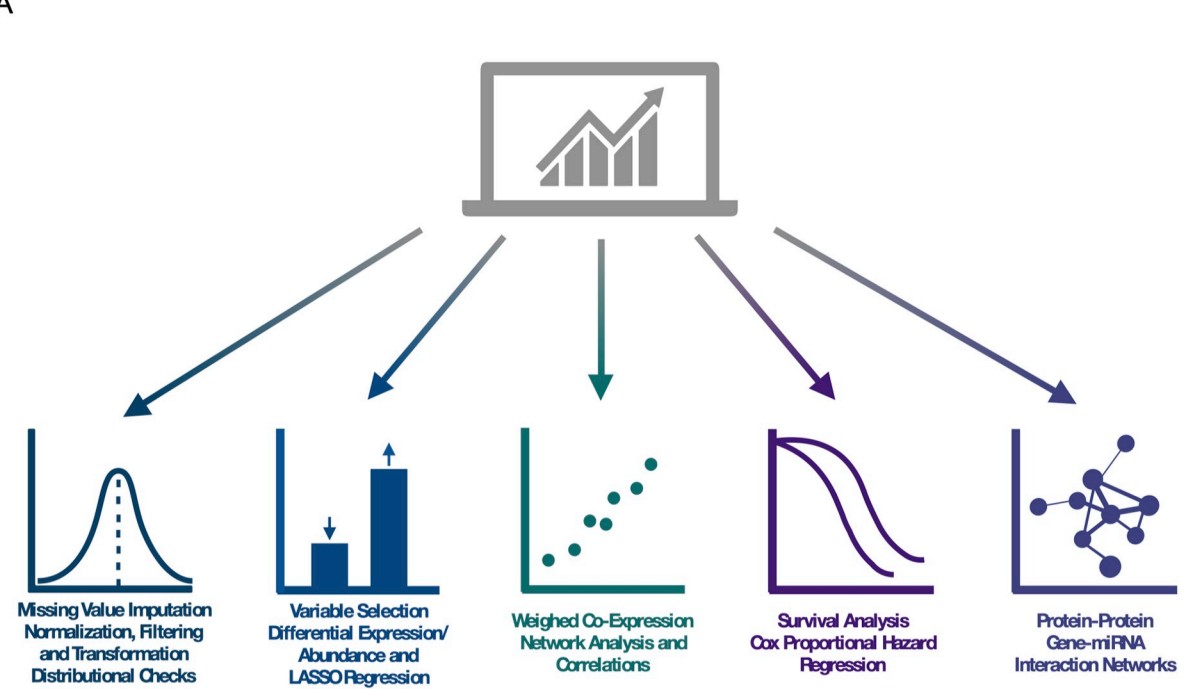

B

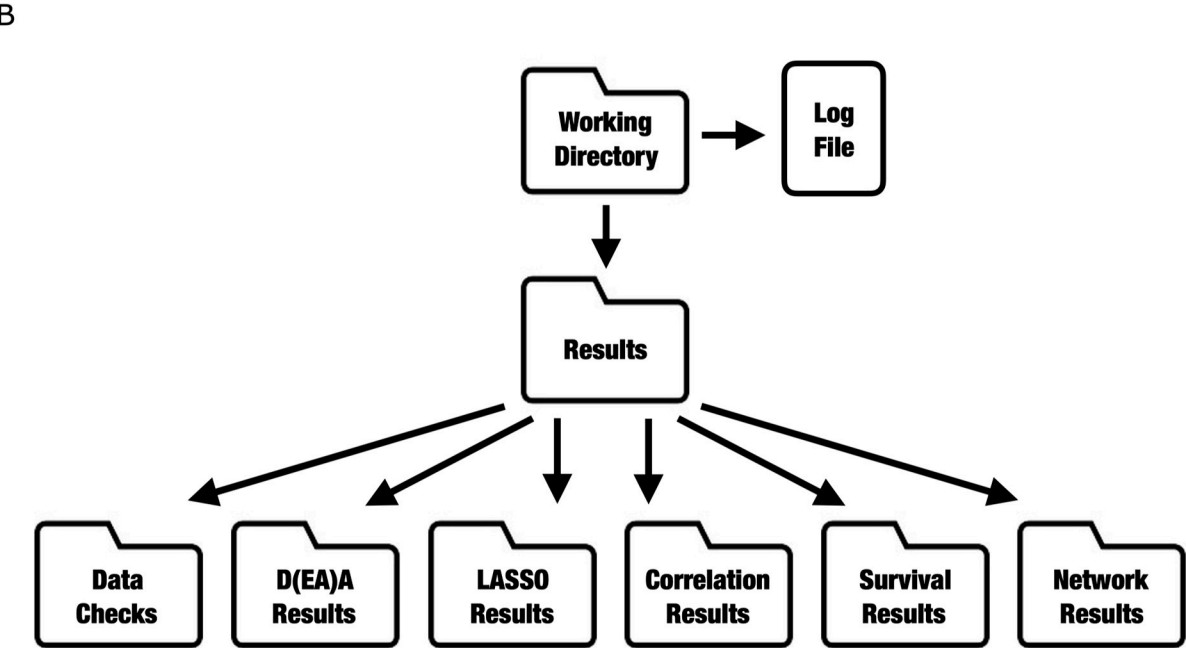

**Fig 1.  A.** Diagram depicting the different types of methodologies and analyses employed by the CAncer bioMarker Prediction Pipeline (CAMPP). **B.** Diagram depicting the structure of CAMPP output, with folders and subfolders organized by analyses.

## Preliminary data distributional checks

As default, the pipeline utilized the R-package *fitdistrplus* [35] to generate skewness-kurtosis plots (Cullen and Frey graphs) for ten randomly selected variables. Distributions are fitted to data by maximum likelihood, and parameters of the distribution are estimated with bootstrap-resampling to simulate variability [35]. In addition to the Cullen and Frey graphs, histograms, quantile-quantile, and probability-probability plots are returned.

## Clustering of samples

CAMPP will test several of centroids for K-means clustering, the exact number of which will depend on the size of the dataset. After clustering, the R-package *mclust* [36] is employed to evaluate which number of clusters is "optimal" for the input data, based on the Bayesian Information Criterion (BIC). The pipeline will return an Multidimensional Scaling (MDS) plot depicting the "best" clustering of samples. In addition, CAMPP will output the original meta-data file with a column specifying which cluster each sample was assigned to. If desired, the user may re-run CAMPP, using the k-means column for variable selection.

## Variable selection with differential expression analysis and elastic-net regression

Variable selection with CAMPP employs *limma* (linear models for microarray data) for differential expression/abundance analysis (DEA, DAA) [37]. *Limma* was originally designed for analysis of microarray data and subsequently revised to handle RNA sequencing data. However, this software is very flexible and has recently been shown to also perform very well with quantitative mass spectrometry data [38,39]. In addition to being versatile, *limma* has been shown to work exceptionally well on datasets with small sample sizes [17,40]. DEA may be performed with correlation for experimental batches and other confounders. Batch-correction is performed by directly incorporating batches into the model design matrix [41]. Batch correction is achieved by specifying the name of the column in the metadata file, which contains the batch information (with flag *-b*)—see user manual for specifics.

As DEA most often yields a long list of variables, CAMPP also performs Least Absolute Shrinkage and Selection Operator (LASSO) or Elastic Net (EN) regression with *glmnet* [42].

EN/LASSO may be performed in two ways; (I) the dataset is split into training and testing subsets, k-fold cross-validation is performed on the training dataset, followed by estimation of specificity and sensitivity (area under the curve = AUC) [43] using the test dataset, or (Il) k-fold cross validation is performed using the full dataset. CAMPP will automatically estimate whether the input dataset is large enough to split. The pipeline will perform regression analysis ten times and output bar-plots of cross-validation errors and AUCs for each run. Results of DEA, LASSO/EN regression, and the overlap between these are output in tables.

## Weighted co-expression network analysis

CAMPP may be used to perform Spearman correlation analysis, with testing for significance and correction for multiple testing (FDR). The user may perform a Weighted Gene Co-Expression Network Analysis. For this type of analysis, CAMPP relies on the R-package WGCNA [44]. To reduce the contribution from low correlations, mainly assumed to be noise, the WGCNA software estimates soft thresholding powers for exponentiation. Co-expression analysis will result in a plot of variable clustering, before merging and after merging of modules (modules with < 25% dissimilarity are merged by default). A heatmap showing the strength of variable co-expression within each module will be generated, if the module

contains $< = 100$ variables—more than this will yield an unreadable plot. CAMPP will return tabular .txt files, one from each module, with the topmost interconnected variables within a module (default is 25%) and accompanying interconnectivity score plots.

### Survival analysis—Pinpointing prognostic biomarkers

Users may perform survival analysis with Cox proportional hazard regression [45] within CAMPP. To run survival analysis, the provided metadata file must contain at least three columns; *age* = age in years at diagnosis, surgery, or entry into trial, *outcome.time* = time until the end of follow-up and *outcome* = specifying censuring or death (0 or 1). In addition to age, the user wishes to correct for other potential confounders. The pipeline checks two underlying assumptions of the Cox model before performing survival analysis: (I) a linear relationship of continuous covariates with log hazards, and (II) proportional hazards of categorical and continuous covariates, i.e., constant relative hazard [46]. If the requirement of linearity is not fulfilled, cubic splines will be added to the covariate(s) in question.

### Interaction networks

After variable selection, the user may generate protein-protein and/or miRNA-gene interaction networks. If gene expression data are used as input for CAMPP, protein-protein interactions are retrieved from the STRING database [47], and pairs, where both genes (proteins) are differentially expressed, are extracted. The pipeline can accept a variety of gene identifiers. If miRNA expression data are used as input, then miRNA-gene interaction pairs are retrieved from either miRTarBase (validated targets) [48], TargetScan (predicted) [49], or a combination of both [50]. Mature miRNA identifiers or miRNA accession are allowed as input. If the user has both gene and miRNA expression values from the same sample cohort, both protein-protein and miRNA-gene pairs are retrieved, and the results are combined. In this case, the pipeline will return pairs where the fold changes of gene and miRNA are inverse, one up-regulated and the other down-regulated. Interaction network analysis with CAMPP will result in a tabular .txt file with all extracted interactions, including logFCs, FDRs, and interaction scores. This file may be used for visualization of networks with Cytoscape [51] or another similar tool. In addition, a plot of the top 100 "strongest" interactions are returned.

## Results

### Case Study 1

Analysis of Single-Channel Microarray Data, Variable Selection, WGCNA and, Gene-Gene Interactions.

For the testing of an array dataset, we used mRNA expression data from single-channel microarrays [52]. The dataset contained 80 breast tumor samples with expression quantified for ~ 15.000 mRNAs. Clinical data included, among other things, information on the classification in breast cancer subtypes. Data were background-corrected for ambient intensities before analysis.

We used CAMPP to perform variable selection. Missing values were imputed, data were log2 transformed and normalized between arrays. Additionally, data were corrected for experimental batches and tumor immune scores. Variables with the ability to separate patient estrogen receptor status (ER+ vs. ER-) were selected with elastic-net (alpha = 0.5) and DEA. For this contrast, patient distribution across the two groups was balanced, and CAMPP divided the data into training and testing sets. Subtype-specific expression profiles were also compared; however, as only nine samples where available for one of the subtypes, elastic-net regression

was performed without splitting the dataset. As the lack of a test set increases the chance of overfitting significantly, elastic-net results merely provided support for DEA results.

Fig 2 shows an example of the data checks performed with *fitdistrplus* [35] on a set of n (default is 10) randomly extracted variables from the dataset. The gene used as an example in Fig 2 is FAM27E2 (Family With Sequence Similarity 27 Member E2), randomly selected from the ten data check plots. FAM27E2 and the other nine genes tested by random, all displayed approximate normal distributions, indicating that these were appropriate for further analysis (see specifics on this in user manual in the GitHub repository). Fig 3 shows the number of up- and down-regulated genes from DEA overlapped with results of elastic-net regression, for the comparison of estrogen receptor status (ER- vs. ER+). Fig 3 also contains a MDS plot for data-overview and statistics on cross-validation error and area under the curve (AUCs) for each of the ten elastic-net runs. Lists of DE genes, results of LASSO regression, and all plots may be found in the *GitHub repository (CS1.zip)*.

As seen from Fig 3B, a total of 147 genes were found to be down-regulated in ER- vs ER+ (up in ER+), while 156 genes were up-regulated in ER- vs. ER+ (down in ER+). Elastic-Net regression resulted in 20 genes, out of which nine overlapped with DEA results. Plots of cross-validation errors and AUCs displayed good convergence, with AUCs ranging between 0.96–1.0. The inspection of the nine genes from the overlap between DEA and elastic-net revealed that two of these were Estrogen Receptor 1 and RAS Like Estrogen Regulated Growth Inhibitor, both up-regulated in the ER+ samples compared to ER- samples, in accordance with expectation [53]. Two of the genes which were up-regulated in ER-negative samples encoded for solute carriers, known to be associated with more aggressive types of breast cancer [54–56]. The heatmap in Fig 4 shows the partitioning of breast tissue samples based on the consensus set of variables from DE analysis and elastic-net regression.

In summary, a total of 290 genes were differentially expressed between subtypes, out of which 20 were also identified by elastic-net regression—in total elastic-net returned 42 genes. The set of 20 consensus genes encompassed many well-known genes with the potential to distinguish subtypes. In addition to three Pam50 genes; ERBB2, ESR1 and FOXA1 [57], the consensus set included; C1orf64 (ER-related factor, ERRF) which was down-regulated in TNBC, in accordance with literature [58,59], CDK12, which was up-regulated in Her2 samples, supported by literature indicating that a total of 71% of Her2-enriched tumors overexpress this gene [60]. Other genes of interest were CYP2B6 [61], MYLK3 [62], and SLURP1 [63].

We used CAMPP to perform WGCNA. In this example, WGCNA was generated only for the DE genes. Fig 5 shows a subset of results from WGCNA with genes DE between breast cancer subtypes. Fig 5 sub-Figs; (I) 5A, a module dendrogram, (II) 5B, an example of a module heatmap showing co-expression of genes in one of the small modules, module 2, and (III) 5C, an example of a module interconnectivity plot, for the top 25% (default setting) most interconnected genes in module 2. The six genes returned as the most interconnected in this module were strongly associated with the Her2-enriched BC subtype. These genes included ERBB2 (HER2 itself), GRB7 (a Pam50 classifier gene) and MIEN1, CDK12, PGAP3, and TCAP, all of which are ERBB2 amplicon passenger genes [64–66].

Lastly, to evaluate whether the variables found to partition BC subtypes were predicted to interact, we used the pipeline to generate protein-protein interaction networks. Fig 6 shows the top 100 strongest gene-gene interactions based on absolute logFC, and interaction score for the comparison of Her2-enriched vs Luminal A, as an example. Results for comparison of all subtypes pairwise may be found in the *GitHub repository (CS1.zip)*. From the plot in Fig 6, the two most interconnected genes in the contrast of Her2-enriched vs Luminal A samples, were ERBB2 and ESR1 (Estrogen Receptor 1). As expected, ERBB2 was highly up-regulated,

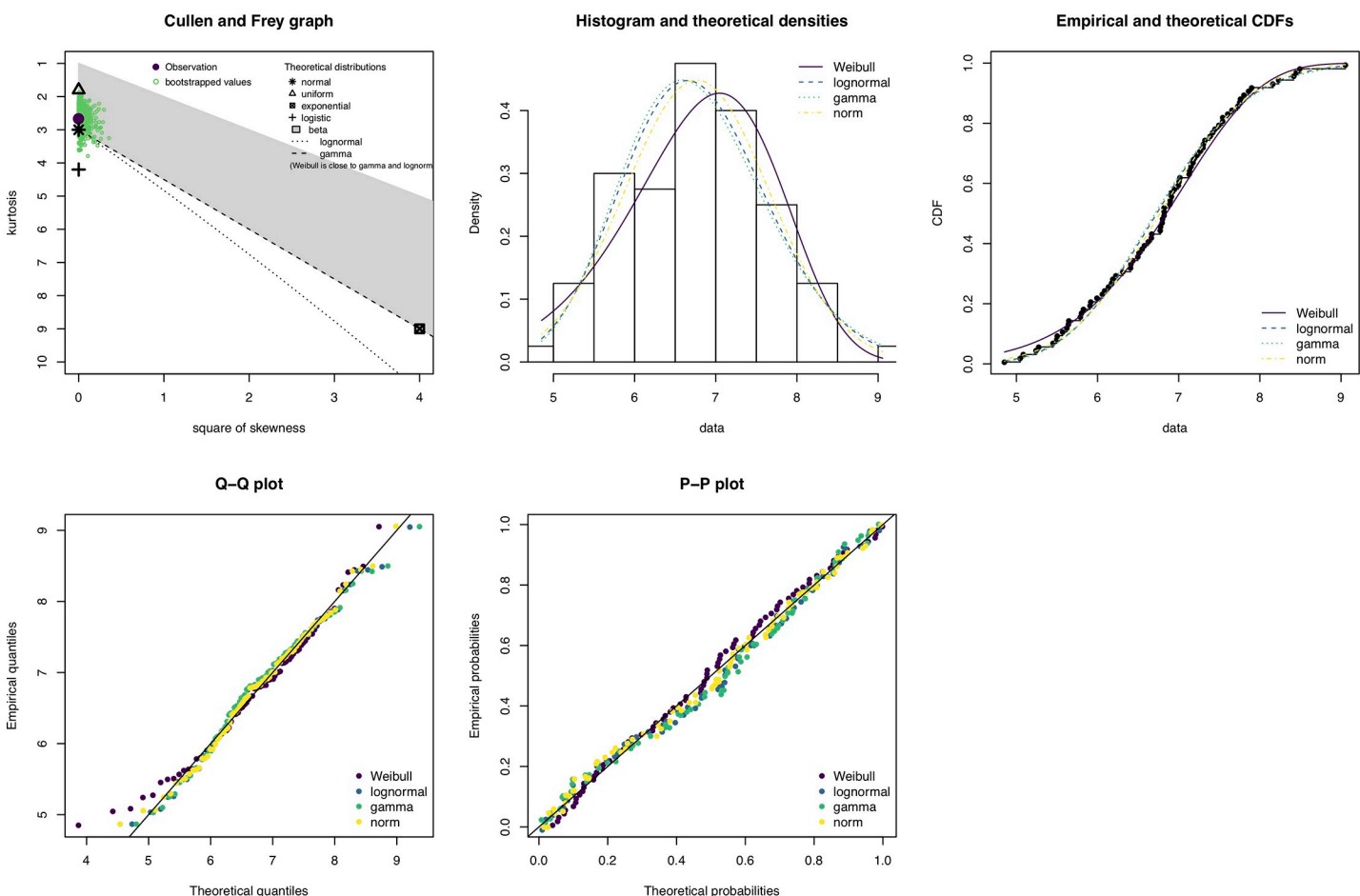

**Fig 2. The output of a CAMPP data check.** The gene used in this example is FAM27E2, randomly selected from the ten variable check plots. Top panel from the left; Cullen and Frey graph showing skewness and kurtosis of normalized and transformed expression data and histogram with different distribution models overlayed. Lower panel from left, quantile-quantile, and probability-probability plot.

while ESR1 was down-regulated in this comparison. Collectively, results highlight the utility of the complementary R-frameworks implemented in the CAMPP wrapper.

## Case Study 2

Analysis of N-glycans from LC Tandem Mass Spectrometry—Clustering, Tissue-Serum Correlation and, Survival Analysis

For case study 2, we analyzed quantitative N-glycan data from liquid-chromatography tandem mass spectrometry (LC-MS/MS). N-glycan abundances have been quantified from tumor and normal interstitial fluids (TIFs and NIFs). In addition to interstitial fluids, we had paired serum samples, enabling us to perform TIF-serum N-glycan abundance correlation analysis with CAMPP. We have published the results of the in-depth analysis of these data here; [67]. Before differential abundance analysis, we used the pipeline to perform K-means clustering of the samples. When running K-means with CAMPP, the user may specify labels to add to the MDS plots generated, to see which clinical variables, if any, best explain the observed clusters. From the evaluation of all k-means plot, the best number of clusters appeared to be two, one corresponding to tumor samples and one to normal samples. S1 Fig shows the two clusters with labels.

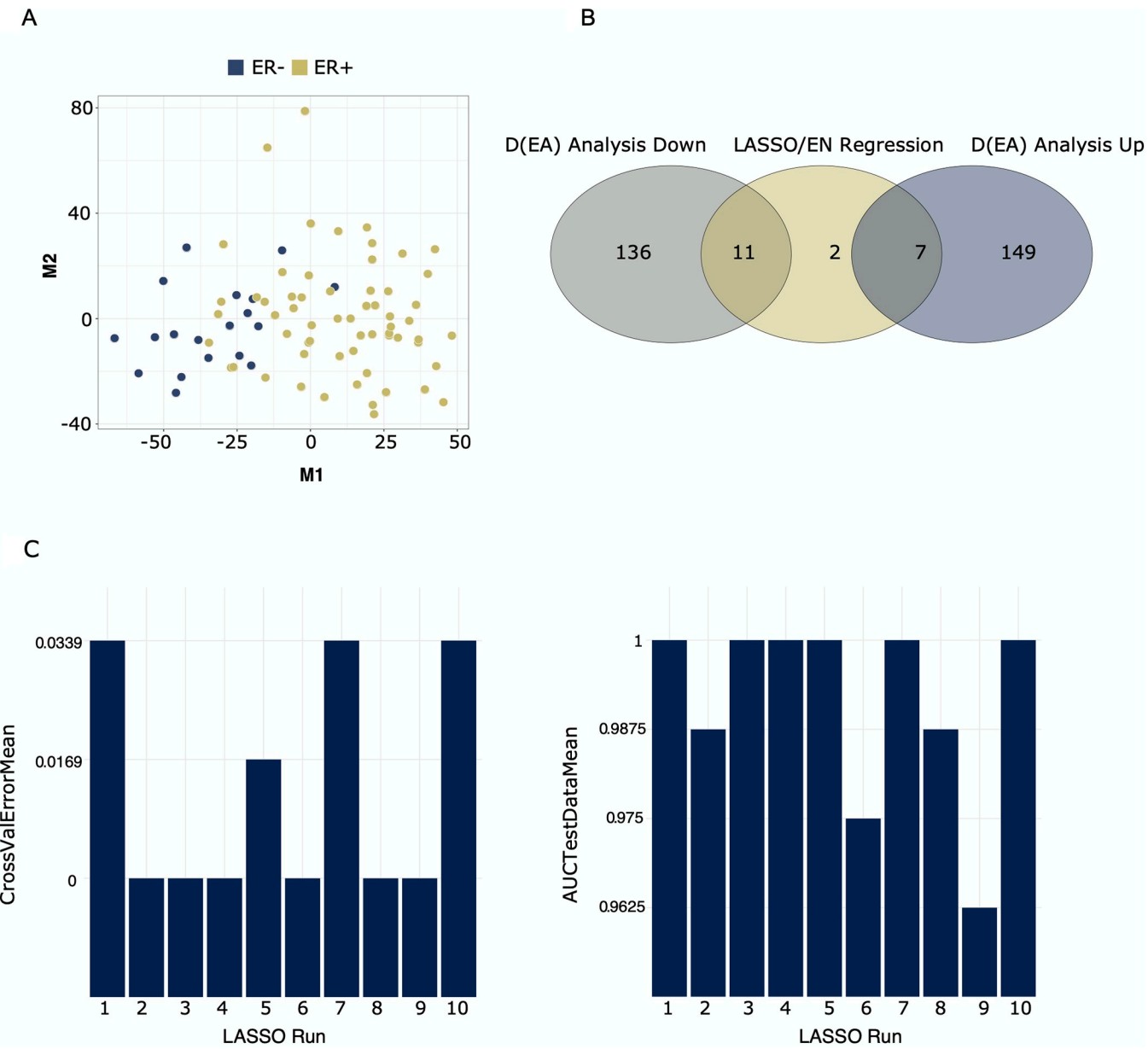

**Fig 3. Results of gene selection using DEA and elastic-net regression.** The dataset contained ~ 15.000 genes and 80 samples, groups used for contrast were estrogen positive (n = 61) vs estrogen negative samples (n = 19). Fig 3A is a multidimensional scaling plot showing the partitioning of samples (based on all genes), colored by estrogen status. Fig 3B shows the overlap of results from elastic-net regression (alpha = 0.5) and differential expression analysis with significance cutoffs logFC > 1 or < -1 and FDR < 0.05. Fig 3C depicts the performance statistics for elastic-net regression, e.g., 10-fold cross-validation errors and area under the curve (AUC) scores for the test set. Elastic-net is run 10 times with different random seeds.

After clustering analysis, we performed DAA and elastic-net regression, as described in the section on variable selection and in Case Study 1. Cut-off for FDR < 0.05 and for elastic-net regression alpha was set to = 0.5. DAA yielded a total of 20 N-glycan groups (12 up-regulated in cancer vs normal and eight down-regulated in cancer vs. normal), while elastic-net returned six N-glycan groups, all encompassed by the DA set—results may be found in the GitHub repository *(i.e., CS2.zip)*. Serum correlation analysis and survival analysis were performed on the set of DE N-glycans. Fig 7 shows the results of the correlation analysis with paired N-glycan

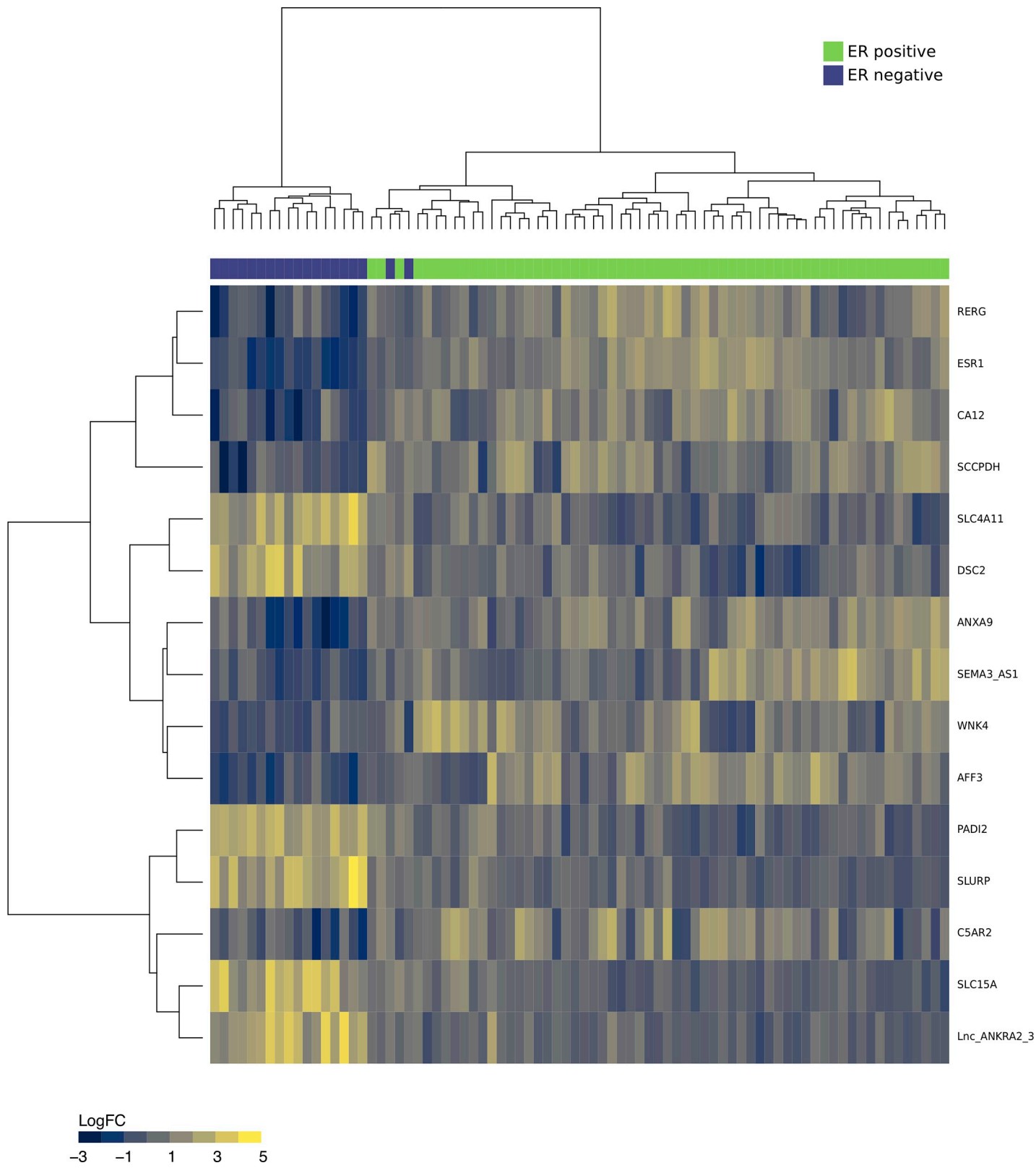

**Fig 4. The heatmap in Fig 4 shows the partitioning breast cancer tissues into estrogen receptor-positive (ER+) samples and estrogen receptor-negative (ER-) samples, based on the consensus set of variables from differential expression analysis and elastic-net regression.** Green = ER+ samples and Purple = ER- samples. Color scale of heatmap (blue to yellow) denotes log2 fold change.

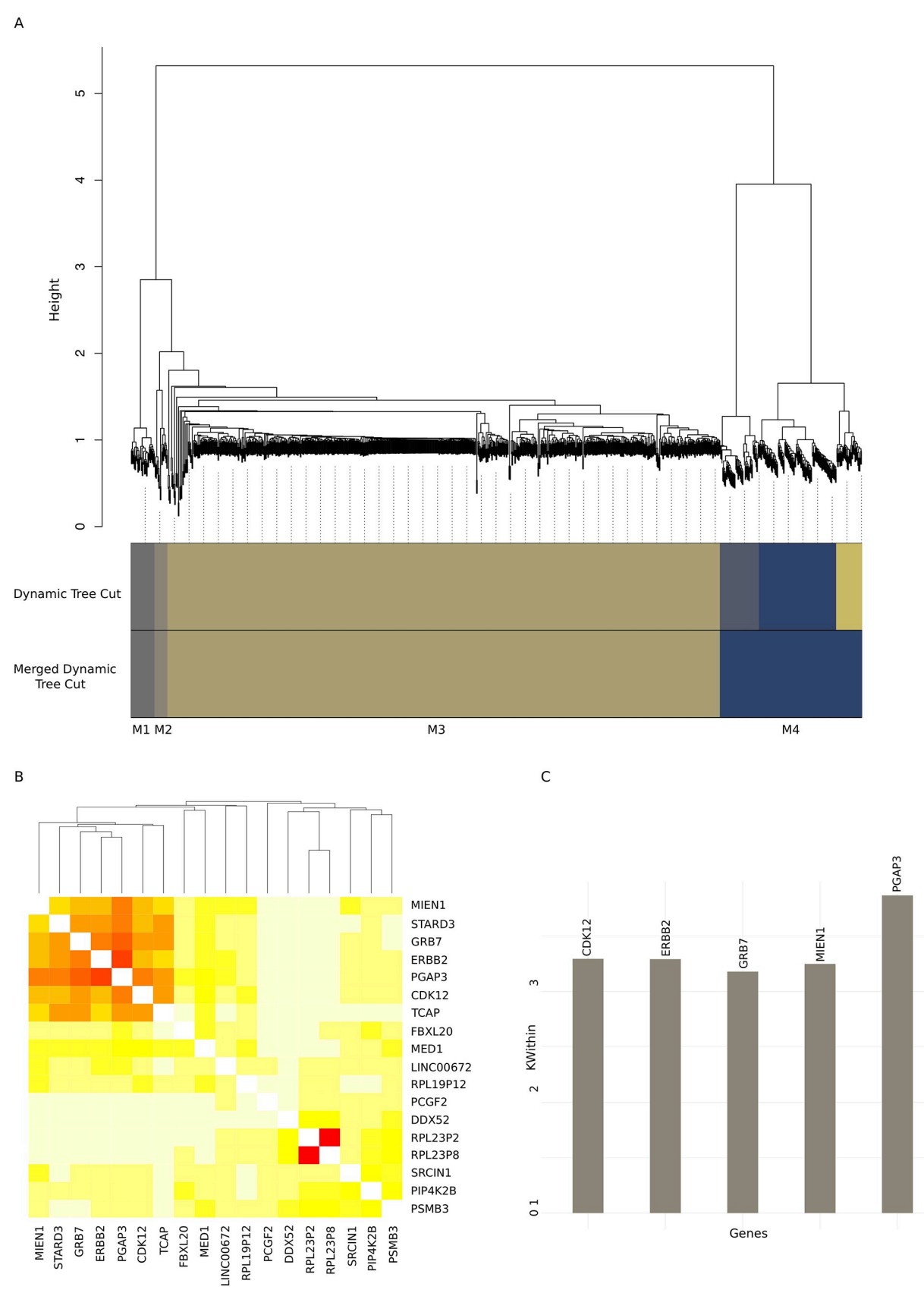

**Fig 5. Results of Weighted Gene Co-expression Network Analysis on dataset of ~ 15.000 genes and 80 samples.** As the dataset contained more than 5000 variables, WGCNA was performed in a block-wise manner to save computational time, in accordance with the WGCNA reference manual [44]. Fig 5A shows the module clustering tree for the first block as an example. Fig 5B depicts the co-expression heatmap for the small module 2, in which a set of six genes display highly correlated expression patterns. Fig 5C contains the top 25% (in this case five) most interconnected genes from the small module 2, with module interconnectivity scores.

abundances from TIFs and serum. Three N-glycan groups, GP1, GP37, and GP38 were found to displayed significant correlation scores after correction for multiple testing. Fig 7A shows correlation scores of all tested N-glycans, while 7B shows the individual scatter plots produced for the three significant N-glycan groups. Lastly, Cox proportional hazard regression was performed with correction for age at diagnosis as well as immune scores, as immune infiltration has been shown to be associated with patient response to treatment and survival [68]. Fig 8 shows the hazard ratios for the DA N-glycan groups with confidence intervals. As seen from

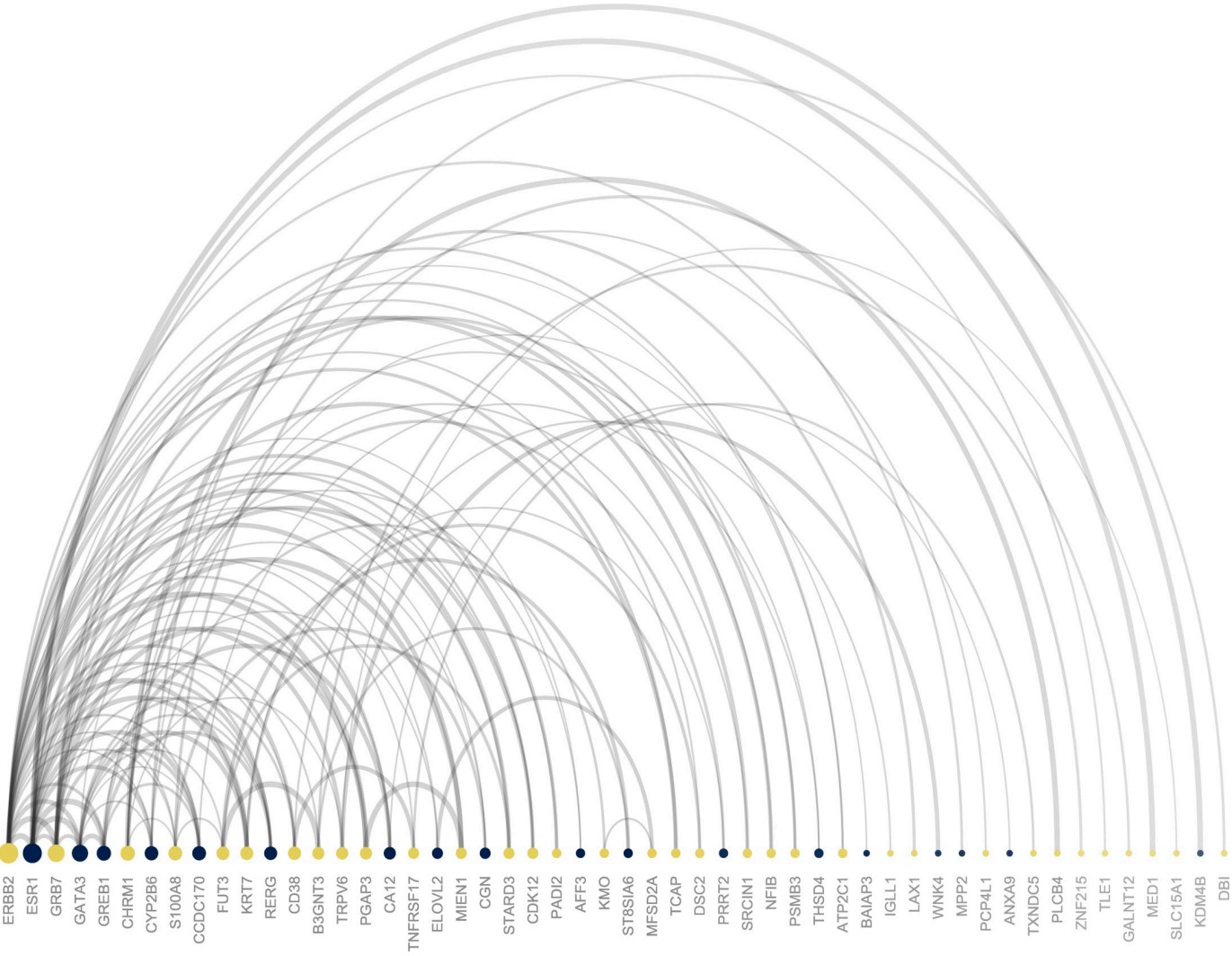

**Fig 6. Plot showing the top 100 best protein-protein (gene-gene) interaction pairs from the analysis of HER2-enriched vs Luminal A samples.** Colors denote the log fold change of a gene; yellow = up-regulated and blue = down-regulated. The size of the node shows the absolute log fold change, while the ordering from left to right denotes the degree of node interconnectivity. The width of the arch represents the interaction score from the STRING database.

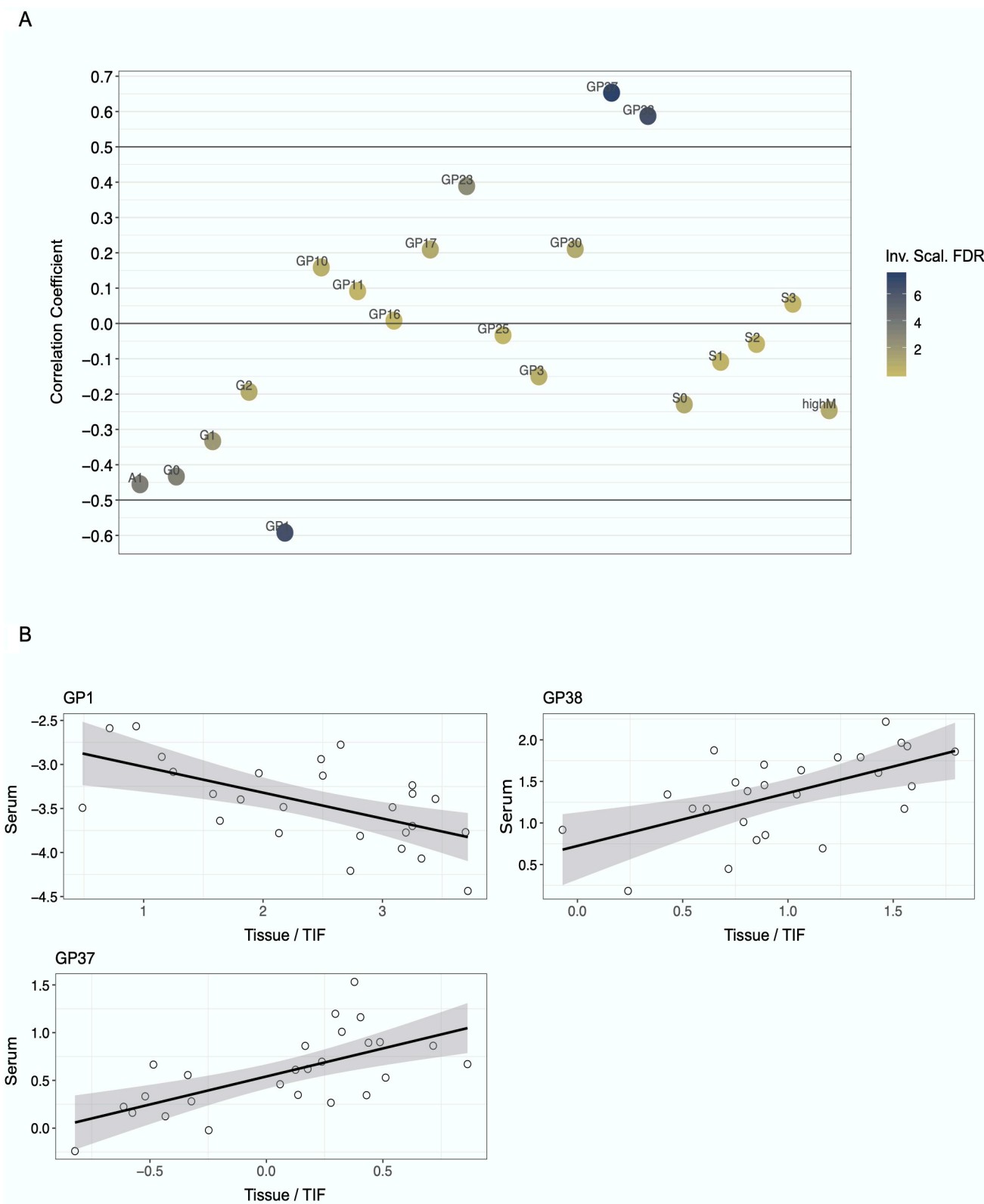

**Fig 7. Results of correlation analysis with N-glycan abundances in interstitial fluids and paired serum samples.** Dataset contained a total of 103 samples (51 normal interstitial fluids and 52 tumor interstitial fluids) with ~70 N-glycan groups (165 N-glycans). Fig 7A shows the correlation scores for differentially abundant N-glycan groups, three of these, GP1, GP37, and GP38 met the requirement for significance (corr > 0.5 and fdr < 0.05), y-

axis = Spearman correlation coefficient. Fig 7B shows the individual correlation plots for the three significant N-glycan groups, x-axis = tumor interstitial fluid abundance and y-axis = serum abundance.

the figure, one N-glycan group, GP38, was significantly associated with overall survival, e.g., a high level of TIF GP38 was predictive of a poor patient outcome.

## Discussion

The CAMPP pipeline supports different types of analysis and will provide the user with graphics to support results—all plots displayed in this publication were generated with CAMPP with no or very minimal editing. CAMPP is implemented in such a way that a user is able to run the pipeline on their local computer on datasets with up to 300–500 samples and 40.000 variables. As an example of this, the user manual, (https://github.com/ELELAB/CAncer-bioMarker-Prediction-Pipeline-CAMPP/blob/master/CAMPPManual.pdf) includes the analysis of breast cancer RNA-seq data from TCGA (416 samples and ~ 32.000 gene transcripts, after filtering). For larger datasets, CAMPP may become slow and memory consuming if WGCNA is performed. In this case, it is advisable to run the pipeline on a server, allocating more cores—Table 2 shows run time, and memory use by CAMPP applied to datasets of different sizes.

We compared the pipeline to other open-source tools for throughput data analysis, found to have similar functionalities and user demographics. S1 Table shows a summary of the CAncer bioMarker Prediction Pipeline (CAMPP), alongside a selection of other software [69–79]. As seen from this comparison the strengths of CAMPP lie both in (I) the variety of analysis it can perform, (II) that the pipeline is able to handle different types of quantitative biological

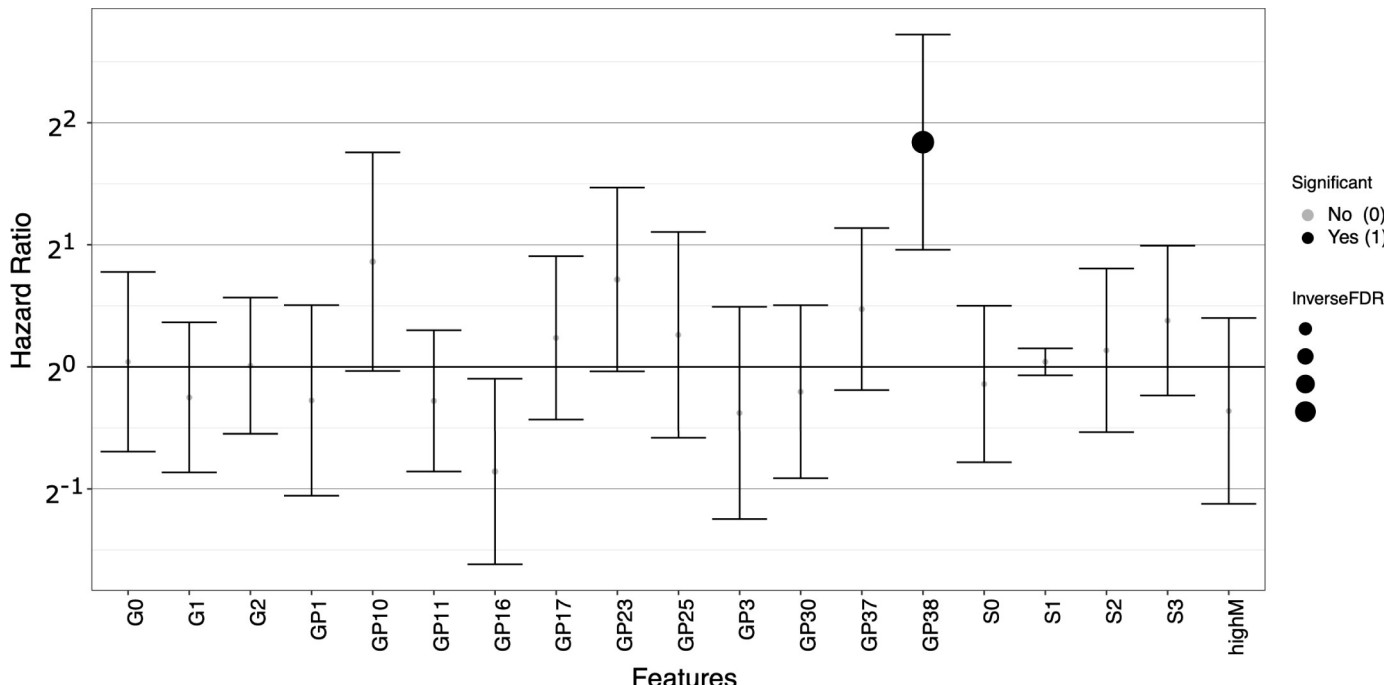

**Fig 8. Results of survival analysis (cox-proportional hazard regression) with correction for patient age at diagnosis and tumor infiltrating lymphocyte status (TILs).** Survival analysis was run on the set of differentially expressed N-glycan groups. Only one N-glycan, GP38, was significant after correction for multiple testing. Hazard ratios are displayed on a log2 scale with confidence intervals, x-axis = N-glycan groups, and y-axis = log2 hazard ratio.

**Table 2. Table showing run times and memory usage for CAMPP applied to datasets of different sizes.** As the weighted gene co-expression network analysis (WGCNA) and estimation of optimal number of clusters for k-means are by far the slowest and most memory consuming processes, we have provided estimates with and without these two analyses. The [.] denotes that a given analysis was not performed on a dataset.

| | Type of Data | Biological Type | Number of Samples | Number of Variables | WGCNA Variables | K-means | Interaction Networks | Run Time in Minutes | Memory Usage in GB |
|---|---|---|---|---|---|---|---|---|---|
| Dataset 1 | Mass Spectrometry | N-glycans | 80 | 165 | All | Yes | . | 0,3 | 0,6 |
| Dataset 2 | Array | mRNA | 80 | 15000 | . | . | . | 0,8 | 0,8 |
| Dataset 2 | Array | mRNA | 80 | 15000 | Differentially Expressed | Yes | . | 3,3 | 1,4 |
| Dataset 2 | Array | mRNA | 80 | 15000 | All | . | . | 12 | 7,8 |
| Dataset 2 | Array | mRNA | 80 | 15000 | All | Yes | . | 15,5 | 7,8 |
| Dataset 3 | Array + Array | microRNA + mRNA | 80 | 15754 | . | . | Yes | 4 | 1,6 |
| Dataset 3 | Array + Array | microRNA + mRNA | 80 | 15754 | Differentially Expressed | Yes | Yes | 5,6 | 1,6 |
| Dataset 3 | Array + Array | microRNA + mRNA | 80 | 15754 | All | . | Yes | 16 | 7,8 |
| Dataset 4 | Array | mRNA | 80 | 29274 | . | . | . | 1,2 | 0,9 |
| Dataset 4 | Array | mRNA | 80 | 29274 | Differentially Expressed | Yes | . | 3,8 | 1,9 |
| Dataset 5 | Sequencing | mRNA | 416 | 55150 | . | . | . | 3 | 1,5 |
| Dataset 5 | Sequencing | mRNA | 416 | 55150 | Differentially Expressed | . | . | 3,5 | 1,4 |
| Dataset 5 | Sequencing | mRNA | 416 | 55150 | Differentially Expressed | Yes | . | 12,8 | 2,3 |

data, from different platforms and (III) that it is flexible in terms of modeling co-variates. Also, unlike most of the other tools, CAMPP employs *limma* for DE analysis. One advantage of *limma* is that this software has been shown to perform well even with very small sample sizes [17,40,80], a low power scenario that is not unfamiliar in biomedical research. Lastly, the pipeline is relatively fast to run even with larger gene expression datasets and somewhat robust to different operating systems as it relies on R/Rstudio, which are continuously maintained to follow system updates.

We have tried to ensure that the software is user-friendly to those without much programming experience. However, we aware that this tool requires the user to work in a terminal environment and to have R and command-line tools installed on their computer. As such, this pipeline is targeted towards bioinformatic software users and computational-inclined bio-researchers.

In summary, CAMPP is an R-based command-line tool for downstream analysis of high throughput biological data. CAMPP was developed with the intent to provide the biomedical community with an automated way of screening for biomolecules of interest while ensuring a standardized framework for data normalization and statistical analysis.

## Availability and future directions

The CAncer bioMarker Prediction Pipeline (CAMPP) is open source and may be downloaded from the github repository: https://github.com/ELELAB/CAncer-bioMarker-Prediction-Pipeline-CAMPP.

As it may be of interest for CAMPP users to understand in detail, how the underlying code and functions work, our plan for the future is to include an R-markdown code run-though. This way the code may more easily be copied and modified and improved, hopefully for the benefit of the computational biology community and for the pipeline itself, as code suggestions and revisions are welcome.

## Supporting information

**S1 Fig. Multidimensional scaling plot.** Plot showing the result of k-means clustering with k = 2. The two clusters support the presumed difference between N-glycan abundances in normal interstitial fluid vs tumor interstitial fluid samples.
(PDF)

**S1 Table. Comparison of the CAncer bioMarker Prediction Pipeline (CAMPP) with other tools for high-throughput data analysis.** This table includes a detailed comparison of the analyses and tools implemented in CAMPP with respect to other bioinformatic resources available.
(XLSX)

## Acknowledgments

The authors would like to thank Vendela Rissler and Emiliano Maiani for testing CAMPP and for helping us with debugging. The calculations described in this paper were performed using the DeiC National Life Science Supercomputer at DTU.

## Author Contributions

**Conceptualization:** Thilde Terkelsen, Elena Papaleo.

**Data curation:** Thilde Terkelsen.

**Formal analysis:** Thilde Terkelsen.

**Funding acquisition:** Elena Papaleo.

**Investigation:** Thilde Terkelsen, Elena Papaleo.

**Methodology:** Thilde Terkelsen, Anders Krogh, Elena Papaleo.

**Project administration:** Elena Papaleo.

**Resources:** Elena Papaleo.

**Software:** Thilde Terkelsen.

**Supervision:** Anders Krogh, Elena Papaleo.

**Validation:** Thilde Terkelsen.

**Visualization:** Thilde Terkelsen.

**Writing – original draft:** Thilde Terkelsen, Elena Papaleo.

**Writing – review & editing:** Thilde Terkelsen, Anders Krogh, Elena Papaleo.

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
