## [Editor Report · Decision Letter 0]

1 Aug 2019

Dear Dr Papaleo,

Thanks for your Presubmission Inquiry regarding your manuscript 'CAncer bioMarker Prediction Pipeline (CAMPP) - A standardised framework for the analysis of quantitative biological data.'. Although we think that your study is interesting, we do not think it provides a significant enough advance or broad enough appeal to our readership for us to consider publishing in PLOS Computational Biology. However, we very much appreciate your wish to present your work in an open-access publication and therefore want to alert you to an alternative that you may find attractive. PLOS ONE is a unique swift, high-volume system for the publication of peer-reviewed research from any scientific discipline. PLOS ONE aims to exploit the full potential of the web to make the most of every piece of research; if you would like to submit your work to PLOS ONE, please visit www.plosone.org and submit your work online. 

Editor's specific comments:

Please excuse the delayed response on your presubmission inquiry. Unfortunately, the software does not provide the strength of advance in functionality that we are seeking in manuscripts for the Software section of PLOS Computational Biology. Thanks for considering PLOS and I wish you well on your journey to communicate this work to the scientific community.

Thanks for considering PLOS Computational Biology, and good luck with your work.

Yours sincerely,

Aaron E. Darling

Software Editor

PLOS Computational Biology

---

## [Decision Letter · Decision Letter 1]

1 Oct 2019

Dear Dr Papaleo,

Thank you very much for submitting your manuscript 'CAncer bioMarker Prediction Pipeline (CAMPP) - A standardised framework for the analysis of quantitative biological data.' for review by PLOS Computational Biology. Your manuscript has been fully evaluated by the PLOS Computational Biology editorial team and in this case also by independent peer reviewers. The reviewers appreciated the attention to an important problem, but raised some substantial concerns about the manuscript as it currently stands. While your manuscript cannot be accepted in its present form, we are willing to consider a revised version in which the issues raised by the reviewers have been adequately addressed. We cannot, of course, promise publication at that time.

Sincerely,

Mihaela Pertea

Software Editor

PLOS Computational Biology

Mihaela Pertea

Software Editor

PLOS Computational Biology

[LINK]

Reviewer's Responses to Questions

**Comments to the Authors:**

Reviewer #1: The authors present a new data analysis pipeline for commonly used omics data (gene expression microarray and RNA-seq data, as well as mass-spec data). The pipeline uses well tested and commonly used methods in the field (Limma, glmnet, etc.), which is positive. The statistical assumptions and approaches appear sound, and the pipeline produces biologically meaningful output when applied to a real microarray dataset (ER+/- negative identifies ESR1 as top differentially expressed gene). My key main concern relates to the usability and the target user group of the pipeline. See below for detailed comments:

## Major issues

### Target user group

The authors write that “this pipeline is targeted towards bioinformatic software users and computational-inclined bio-researchers.” My main concern is that computational-inclined researchers will prefer to piece together their own pipeline (even if using exactly the same components as CAMPP) to control and test key parameters in the analysis. Instead, I think the pipeline could be very useful for researchers that have limited computational know-how. However, this group of people may likely need a much more thorough and user-friendly explanation of the output than what CAMPP currently provides. From a quick inspection of the output, CAMPP generally provides very technical plots that will be difficult to interpret for the average user with limited statistical expertise.

### RNA-seq data

Most current gene expression data is generated using RNA-seq. It is unfortunate that the authors do not provide an example and validation using this type of data. The authors could for example compare ER+/- RNA-seq data from TCGA and compare with their microarray results. This would also allow the authors to demonstrate the scalability of their software (e.g. running on 300-500 samples and 40.000 variables).

### Minor

### Data distributional checks

The pipeline runs a set of distributional checks on 10 randomly selected variables. This analysis is summarised in a set of plots for each variable. A key problem here is that this information is not summarised and presented well to the user. Assuming that the average user has limited statistical expertise, it is not clear how they should act on this information. Furthermore, it is not clear how the outcome of these checks affect the downstream analysis pipeline.

## Compute requirements

Information on computational/hardware requirements should be more specific. How much RAM is needed to run 300-500 samples and 40.000 variables? What’s the runtime? Can the software take advantage of multi-core systems?

### Batch effects

The authors write that the pipeline can adjust for batch effects, but it is not clear how the user supplies batch information in the input data.

## Comparison with other pipelines, table 2

The information in table 2 is not easy to digest. Firstly, information for CAMPP is not included(?). Secondly, the table is very large (2-pages), so it’s not easy to compare pipelines.

Reviewer #2: Terkelsen et al. present a useful pipeline for the detection of biomarkers from input data matrices such as gene expression or protein abundance values. It covers various aspects of data processing, including normalization and batch effect correction. The tool is available via github and can easily be installed in R. The code is well structured and sufficiently commented. The user manual guides users through installation and usage. It has an open source license and can thus be modified and used freely. It is suitable as a starting place for biomedical researchers familiarizing themselves with biomarker discovery. I have some suggestions for improvements:

CAMPP depends on R packages which tend to change over time. To avoid that CAMPP quickly becomes unusable I strongly suggest one of the following options:

- build a docker container

- use the R packrat package to bind CAMPP to a static version of its dependencies

Table 2 is very clunky. Could this be solved with a table that has checkmarks when features are implemented in one of the tools?

Minor:

- ElasticNet: the parameter alpha for balancing ridge and lasso regression should be adjustable

- It is not clear to me why all four columns are needed for survival analysis. Age is useful to consider but not strictly necessary to perform survival analysis. Is it used as a confounder?

- typo: trail -> trial

**Have all data underlying the figures and results presented in the manuscript been provided?**

Reviewer #1: Yes

Reviewer #2: Yes

PLOS authors have the option to publish the peer review history of their article (what does this mean?). If published, this will include your full peer review and any attached files.

Reviewer #1: No

Reviewer #2: Yes: Markus List

---

## [Decision Letter · Decision Letter 2]

18 Jan 2020

Dear Prof. Papaleo,

We are pleased to inform you that your manuscript 'CAncer bioMarker Prediction Pipeline (CAMPP) - A standardized framework for the analysis of quantitative biological data' has been provisionally accepted for publication in PLOS Computational Biology.

Before your manuscript can be formally accepted you will need to complete some formatting changes, which you will receive in a follow up email. A member of our team will be in touch within two working days with a set of requests.

Best regards,

Mihaela Pertea

Software Editor

PLOS Computational Biology

Mihaela Pertea

Software Editor

PLOS Computational Biology

Reviewer's Responses to Questions

Comments to the Authors:

Please note here if the review is uploaded as an attachment.

Reviewer #1: The authors have addressed all my concerns. I would like to congratulate the authors on a much improved manuscript. Especially the run time information (table 1), the more user friendly table 2, and the references to additional information in the user manual.

Minor point:

Many important details are provided in the user manual, which is great. I would encourage the authors to reference this manual more frequently in the main manuscript text. For example referencing the new RNA-seq analysis of TCGA breast cancer. The authors could for example reference this analysis in the discussion, and comment on whether the analysis results are generally concordant with the microarray analysis (eg1. ESR1 overexpression in ER+/Luminal tumors, and HER2 over expression in HER2+ tumors).

Reviewer #2: The authors have addressed all of my previous comments.

Have all data underlying the figures and results presented in the manuscript been provided?

Large-scale datasets should be made available via a public repository as described in the 

PLOS Computational Biology

data availability policy, and numerical data that underlies graphs or summary statistics should be provided in spreadsheet form as supporting information.

Reviewer #1: Yes

Reviewer #2: Yes

PLOS authors have the option to publish the peer review history of their article (what does this mean?). If published, this will include your full peer review and any attached files.

Do you want your identity to be public for this peer review?

 For information about this choice, including consent withdrawal, please see our Privacy Policy.

Reviewer #1: No

Reviewer #2: Yes: Markus List

---

## [Editor Report · Acceptance letter]

5 Mar 2020

PCOMPBIOL-D-19-01225R2 

CAncer bioMarker Prediction Pipeline (CAMPP) - A standardized framework for the analysis of quantitative biological data

Dear Dr Papaleo,

I am pleased to inform you that your manuscript has been formally accepted for publication in PLOS Computational Biology. Your manuscript is now with our production department and you will be notified of the publication date in due course.

With kind regards,

Matt Lyles
